# A new protocol for multispecies bacterial infections in zebrafish and their monitoring through automated image analysis

**Désirée A. Schmitz**[1,2]*, **Tobias Wechsler**[1], **Hongwei Bran Li**[1,3], **Bjoern H. Menze**[1], **Rolf Kümmerli**[1]*

**1** Department of Quantitative Biomedicine, University of Zurich, Zurich, Switzerland, **2** Department of Microbiology, Harvard Medical School, Boston, Massachusetts, United States of America, **3** Athinoula A. Martinos Center for Biomedical Imaging, Massachusetts General Hospital, Harvard Medical School, Boston, Massachusetts, United States of America

* schmitz.desiree.a@gmail.com (DAS); rolf.kuemmerli@uzh.ch (RK)

**Data Availability Statement:** The survival data is available on the figshare repository (https://figshare.com/account/projects/202542/articles/

## Abstract

The zebrafish *Danio rerio* has become a popular model host to explore disease pathology caused by infectious agents. A main advantage is its transparency at an early age, which enables live imaging of infection dynamics. While multispecies infections are common in patients, the zebrafish model is rarely used to study them, although the model would be ideal for investigating pathogen-pathogen and pathogen-host interactions. This may be due to the absence of an established multispecies infection protocol for a defined organ and the lack of suitable image analysis pipelines for automated image processing. To address these issues, we developed a protocol for establishing and tracking single and multispecies bacterial infections in the inner ear structure (otic vesicle) of the zebrafish by imaging. Subsequently, we generated an image analysis pipeline that involved deep learning for the automated segmentation of the otic vesicle, and scripts for quantifying pathogen frequencies through fluorescence intensity measures. We used *Pseudomonas aeruginosa*, *Acinetobacter baumannii*, and *Klebsiella pneumoniae*, three of the difficult-to-treat ESKAPE pathogens, to show that our infection protocol and image analysis pipeline work both for single pathogens and pairwise pathogen combinations. Thus, our protocols provide a comprehensive toolbox for studying single and multispecies infections in real-time in zebrafish.

## Introduction

Using animal models in research enables us to gain a deeper understanding of infections and the interactions between co-infecting pathogens and pathogens and the host. However, in recent years there have been increased public objections to the use of animals for research purposes [1]. Even though invertebrates like insects (e.g. the larvae of *Galleria mellonella*) or worms (e.g. *Caenorhabditis elegans*) are valid alternative host organisms, they lack many similarities to the human immune response and should thus only be considered as a first step towards a better understanding of human bacterial infections. In contrast, mice and rats are genetically and physiologically much more similar to humans and have become the primary

25669311). All images have been deposited in the BioImage Archive (https://www.ebi.ac.uk/biostudies/bioimages/studies/S-BIAD1159?key=ff1fb1fc-54f1-47fc-b14d-c1bf4c41d33c).

**Funding:** Swiss National Science Foundation (grant no. 31003A_182499 and 310030_212266 to RK). The funders had no role in study design, data collection and analysis, decision to publish, or preparation of the manuscript.

**Competing interests:** The authors have declared that no competing interests exist.

vertebrates used for scientific purposes, making up 95% of all animal research [2]. Yet, they display consciousness and have complex cognitive abilities, raising ethical concerns. Much fewer concerns arise with the zebrafish *D. rerio* due to several reasons. While this animal is also a vertebrate, it is not recognized as an animal up to five days post-fertilization from the perspective of animal research. Only after five days, zebrafish begin to respond to various stimuli, including an optomotor response [3], optokinetic response [4], acoustic startle response [5, 6], thigmotaxis [7–9] and develop a circadian rhythm [10]. Before this age, they only have a relatively simple nervous system and are not expected to experience pain or distress, which is characteristic of more mature developmental stages of vertebrates [11]. Nevertheless, they already have innate immunity [12–14]. Moreover, the zebrafish shares more than 80% of human genes associated with disease and its genome has been fully sequenced, making it a valuable system for studying pathogenesis caused by human bacterial pathogens [15–17]. Taken together, the zebrafish model (up to five days post-fertilization) is a good compromise for research because it minimizes pain and distress for animals, while still enabling to work with a relevant vertebrate system.

One of the main reasons why the zebrafish *D. rerio* has become a popular model host to study bacterial pathogenesis is its transparency at an early age [18–20]. This allows for live imaging of infection dynamics locally as well as in the whole zebrafish. Thanks to this visualization, the zebrafish host would be particularly well suited to examine multispecies infections and the pathogen-pathogen and pathogen-host interactions therein. However, so far only single infections in defined organs have been reported and the subsequent analyses were done manually [21–24].

Here, we present protocols that allow to study multispecies infections in zebrafish in an automated fashion in a defined organ so that the frequency and behavior of pathogens can be monitored over time. Our three-stage protocol describes (A) single and multispecies infections into the inner ear structure of zebrafish, the otic vesicle, (B) embedding zebrafish and imaging the otic vesicle, and (C) automated segmentation of the otic vesicle and subsequent image analysis. This workflow allows imaging of infection dynamics and their consequences from both the host and the pathogen side. For example, zebrafish lines with fluorescently tagged macrophages can illuminate the host's immune response and fluorescently tagged pathogens let us explore pathogen-pathogen and pathogen-host interactions. Furthermore, the use of gene reporter strains allows investigating the regulation of specific pathogen genes and their importance within an infection.

To establish our infection protocol, we used fluorescently tagged *K. pneumoniae*, *A. baumannii*, and *P. aeruginosa* strains. These three opportunistic human pathogens belong to the so-called ESKAPE pathogens, which stands for six bacterial species that are of particular concern regarding infections because they are typically multidrug-resistant, highly virulent, and co-occur in several polymicrobial infections [25, 26]. Interestingly, the 'KAP' pathogens, *K. pneumoniae*, *A. baumannii*, and *P. aeruginosa*, can be present alone or together in any possible combination in infections, for example in ventilator-associated pneumonia [27]. Thus, there is great interest in understanding how these species affect host morbidity and mortality alone and in combination.

To track co-infections in a living host, we developed an image analysis pipeline to have a solution that is tailored to a specific organ of the zebrafish (the otic vesicle) and can easily be used by other researchers. This has several advantages over other software, like the commercial software Athena for zebrafish segmentation from the company IDEA Bio-Medical [28]. First and most importantly, our solution is completely free of charge and does not require purchasing each image analyzed, which can get expensive quickly with large sample sizes. Second, our segmentation does not need the entire zebrafish to be imaged, which would be time-

consuming during image acquisition and can lead to additional costs when using microscopes from core facilities that are paid for on an hourly basis. Third, we chose the inner ear structure of the zebrafish (otic vesicle) to study infections because the organ is clearly delineated from the surrounding tissue, such that infections can be controlled and species interactions can be monitored over time. We are confident that this combined protocol enables many researchers to investigate infection dynamics and thus advance our understanding of disease pathology relevant for vertebrates including humans.

## Materials and methods

The protocols described in this peer-reviewed article are published on protocols.io (dx.doi. org/10.17504/protocols.io.rm7vzjybxlx1/v1) and are also provided as S1 File with this article. We offer three individual protocols: (A) single and multispecies bacterial zebrafish infections into the otic vesicle, (B) embedding and imaging of zebrafish, and (C) image analysis (automated segmentation and subsequent image analysis) (respectively, dx.doi.org/10.17504/ protocols.io.j8nlk8kwwl5r/v1, dx.doi.org/10.17504/protocols.io.14egn6726l5d/v1, dx.doi.org/ 10.17504/protocols.io.bp2l6219dgqe/v1), which are also available for printing as S2–S4 Files, respectively. For the images and image analyses, we followed the community-developed checklists by Schmied et al. [29].

### Bacterial & zebrafish strains

For all infection experiments, we used the following three pathogens, all of which are opportunistic human pathogens and commonly co-occur in infections: *P. aeruginosa* strain PAO1:: mCherry (wildtype from [30]; tagged by the Kümmerli lab), *K. pneumoniae* strain CH1477 GFPmut3 and strain CH1478 mCherry [31], and *A. baumannii* strain AB5075-F sfGFP [32]. All strains were tagged with a constitutively expressed fluorescence marker, inserted at the attTn7 site, which is not associated with any growth defect. The wildtype strain AB of the zebrafish *Danio rerio* was used for all experiments. As per the EU Directive 2010/63/EU concerning the welfare of animals employed in scientific research, the initial life stages of animals are excluded from protection [33, 34]. For zebrafish embryos/larvae, these initial stages include up to 5 days post-fertilization, which is why we ceased all experiments on day 5. Consequently, our work does not fall under the regulatory frameworks governing animal experimentation and no ethics approval is required.

### Mold for injecting 2 days post-fertilization zebrafish

For zebrafish infections, we adapted the protocol by Benard et al. [21] for multispecies infections into the otic vesicle. We used the microstructure surface array designed by Ellett & Irimia [35]. Their mold allows for an improved infection procedure as compared to the more standard mounting of zebrafish in agarose or methylcellulose, which is time-consuming and technically challenging. Specifically, the micro-structured surface array consists of a rubber mold that can be reused for making agarose molds to align 2 days post-fertilization zebrafish larvae in one of three orientations, dorsally, ventrally, and laterally. For injections into the otic vesicle, the channel for lateral orientation can be used to optimally position 10–12 zebrafish.

### Development of the automated segmentation with deep learning

Up to now, regions of interest in zebrafish images had to be defined manually. Here, we have established an automated segmentation procedure for the inner ear structure of the zebrafish (otic vesicle) that excludes the enclosed calcite dots, which inherently fluoresce and cannot be

inhabited by pathogens. Using deep learning algorithms, we developed an automated segmentation tool by implementing the following four steps: (1) preparation of data for training, (2) initial model training and optimization, (3) model re-training with label correction on difficult samples and semi-supervised learning, and (4) model containerization.

1. In the data preparation stage, we manually segmented images in Ilastik [36] using the brightfield channel. Images were then resized from 2048 * 2048 pixels to 512 * 512 pixels to reduce computational complexity. Six individual zebrafish microscopy samples each consisting of several z-stack slices (a total of 67 two-dimensional slices) were manually segmented (using distinct labels for the otic vesicle and the calcite dots) for model training in the next step.

2. In the training and optimization phase we adapted the nnU-Net deep learning framework [37] for supervised training. A 2D deep neural network was trained on the resized images to maximize the match between the automated and the manually conducted segmentation. To account for the fact that images can differ widely in their background, we implemented a background mixup strategy for training [38]. Specifically, we generated a new set of images by taking two batches of randomly sampled images as the input. The mixup process used a convex combination of the backgrounds from these two batches of images. All images and masks were randomly augmented with random rotations, random Gama contrast correction, random shearing, and random scaling to generate diverse training samples. The initial network was trained on an RTX-A6000 GPU for 500 epochs. The best model was picked based on the Dice scores of an internal validation set.

3. The trained network using the above settings is optimized for images of good quality and from a single source (Leica Thunder DMi8 widefield microscope with the Leica monochrome fluorescence DFC9000 GTC camera system). To extend its applicability to images with lower quality and from different sources, we retrained the network in two steps. First, we included images (16 zebrafish samples) that were not perfectly segmented by the initial model (considered as 'hard samples') and manually corrected them. We then re-trained the model with these manually corrected segmentations. In the second step, we applied the re-trained model to images from two different sources (the previously used Leica Thunder DMi8 and in addition the Nikon Eclipse Ti2-E with the camera system Orca Fusion CoaXpress sCMOS). For this final training, we only selected images that were well-segmented. Combining the pool of the original (first training), corrected (first re-training), and the newly selected images (second re-training), the model was trained on a total of 240 2D slides and their segmentations (from 45 individual zebrafish samples). Significant performance improvement was observed when testing a new set of zebrafish samples (see results). We defined a segmentation as successful when at least one z-slice of a zebrafish sample was well segmented. Similar to the data preparation stage, whenever testing on new images, the input images were resized to 512*512 pixels and their output segmentation masks were resized back to the original image size to match the resolution.

4. The final model is available on Docker for free use in the research community: https://hub.docker.com/repository/docker/branhongweili/dqbm_cell_seg/general. It can run on popular operating systems (Ubuntu, MacOS, and Windows 10) using CPU or GPU.

## Time indications for infections, embedding, and imaging

The amount of time needed for infecting zebrafish locally into the otic vesicle (Fig 1, Protocol A), highly depends on the number of treatments applied and the number of zebrafish injected

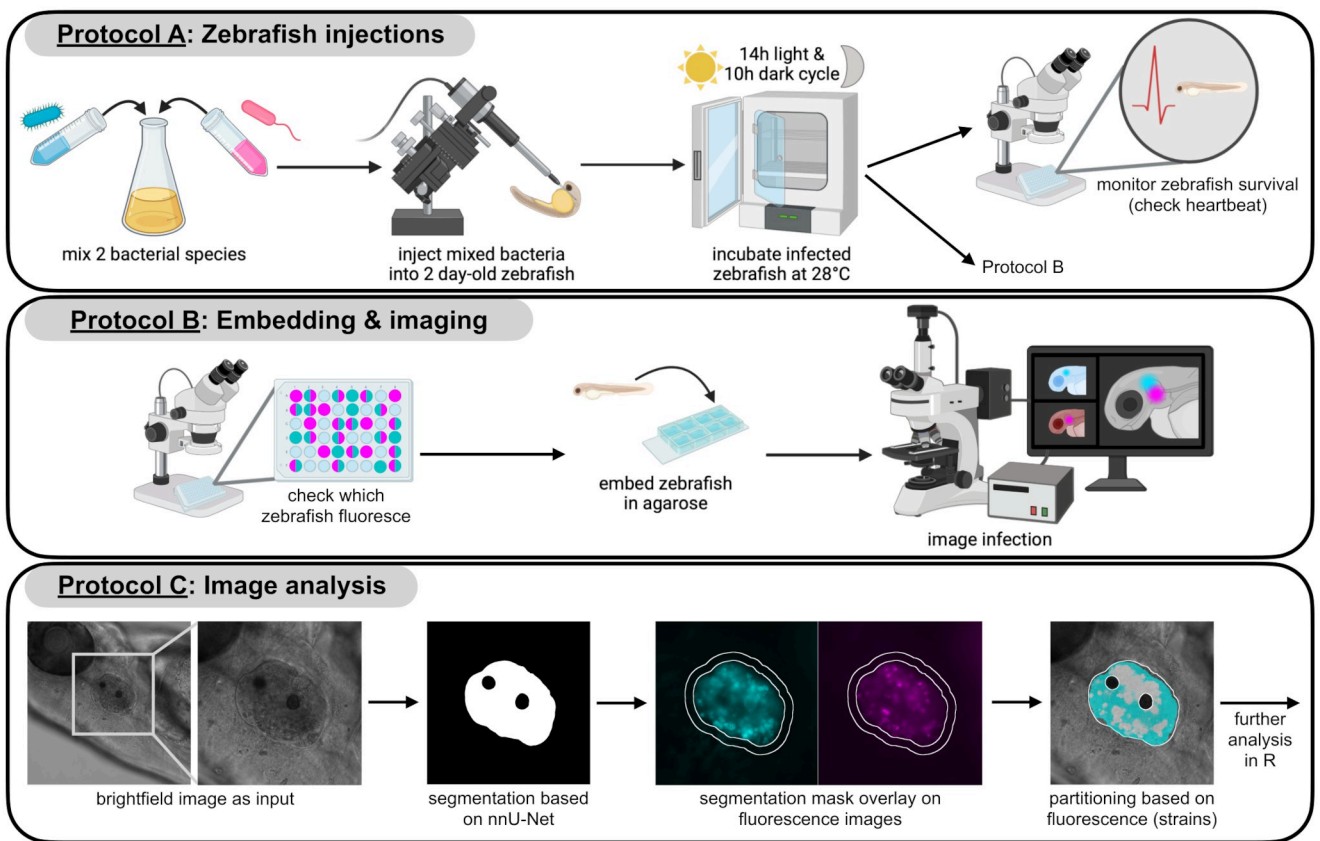

**Fig 1. Overview of the complete workflow, divided into the three protocols.** Protocol A: bacterial species are mixed and injected into the otic vesicle (inner ear structure) of 2 day-old zebrafish, which are then incubated at 28˚C with a 14h-light and 10h-dark cycle (or only in the dark). Zebrafish survival can be monitored with a stereomicroscope by checking the heartbeat. Protocol B: an initial fluorescence screen is performed to selectively choose zebrafish with an ongoing infection for embedding and subsequent imaging. For imaging, zebrafish are embedded in low-melting-point agarose in an 8-well Ibidi μ-slide, adding water on top for proper hydration. An inverted widefield microscope is used for imaging with an objective that has a long working distance. We used widefield instead of confocal microscopy to allow for automated multi-sample imaging. Protocol C: a brightfield image is the input for the automated segmentation to create a mask of the otic vesicle. This mask excludes the calcite dots within the vesicle because they cannot be colonized by bacteria but inherently fluoresce. This mask is then superimposed on the fluorescence images for the quantification of species location and density. See individual protocols for a detailed description.

per treatment. For example, with four treatments (e.g. two single species, one double species, one control injection) and 20–30 zebrafish per treatment, the process takes approximately 4–5 h. Adding another 10–20 zebrafish per treatment will add approximately 15–20 min. Each additional treatment (e.g. different inoculum size) adds about 30–45 min. Less time is required when additional treatments involve the same species as those used before. More time is required when additional species are included. This is because the preparation of additional bacterial suspensions before the injections is time-consuming.

Embedding zebrafish for imaging and the imaging of their otic vesicles (Fig 1, Protocol B) takes approximately 4 h for ~30 zebrafish. The image analysis protocol (Fig 1, Protocol C) including the automated segmentation of the otic vesicle takes around 3 h for ~30 zebrafish (using GPU). For this last protocol, time demand is non-linear, meaning that the time needed per zebrafish declines with more zebrafish samples. Note that all time estimations represent indications that may vary in response to the experimenter's training status and experience.

## Expected results & discussion

This article describes the use of zebrafish for studying single and multispecies infections in a defined organ including automated segmentation followed by quantitative image analysis. It includes the following three protocols: (A) zebrafish infections into their inner ear structure, the otic vesicle, (B) zebrafish embedding and imaging, and (C) automated segmentation of the otic vesicle and image analysis (Fig 1).

### Dose-dependent killing for single infections validates our infection protocol

We infected zebrafish with increasing amounts of *P. aeruginosa* in two independent experiments to demonstrate that our otic vesicle infection protocol works and that host individuals are negatively affected (Fig 2A). We found that host killing was dose-dependent: with significant differences between a low, medium, and high number of bacterial cells injected (log-rank test, alpha = 0.05). Moreover, we observed that mixed infections affected host survival rates. For example, the co-infection of *K. pneumoniae* with *P. aeruginosa* led to a survival probability that lay between the two corresponding mono-infections (Fig 2B). In contrast, for the co-infection with *A. baumannii* and *P. aeruginosa* survival probability was different from the *P. aeruginosa* mono-infection but similar to the *A. baumannii* mono-infection (Fig 2C). These results show that differences between mono- and co-infections can accurately be studied and monitored with our infection protocols. The co-infection of *A. baumannii* with *K. pneumoniae* is shown in the S1 Fig. Note that otic vesicle infections seem to require higher infection doses of *P. aeruginosa* to cause host mortality than other commonly used locations like the yolk circulation valley [39]. One reason could be that the otic vesicle infection is contained in a relatively small organ.

### Reliability of the automated segmentation across experiments and instruments

The complete image analysis procedure is depicted in Fig 3, including all tools and scripts we developed. The calcite dots within the otic vesicle are automatically recognized and removed from the analysis because they cannot be occupied by bacteria but inherently fluoresce. To validate the automated segmentation of the otic vesicle, we used test data from two independent experiments with a total of 57 zebrafish individuals. Our segmentation tool initially worked in about 63–76% of individuals (for at least one z-stack slice). To both increase the success of our segmentation tool and confirm its broad applicability independent of the widefield microscope used for imaging, we conducted a second re-training of our segmentation model (see methods for more details). For this, we obtained images from similar inverted widefield microscopes of two different brands (Leica DMi8 and Nikon Eclipse Ti2-E). We then tested our final segmentation model with images from an additional experiment including approximately 50 new zebrafish samples. We could improve the segmentation success to 85% and 91% for the two microscopes, respectively. Thus, the retraining was beneficial and resulted in a pipeline with high segmentation reliability.

### Two co-infecting species can reliably be detected and quantified in the otic vesicle of zebrafish

We used our scripts to measure the areas occupied by each pathogen in a co-infection using their constitutively expressed fluorescent markers. In most co-infected zebrafish individuals, we detected both bacterial species (Fig 4 and S2 Fig). Moreover, we observed three types of

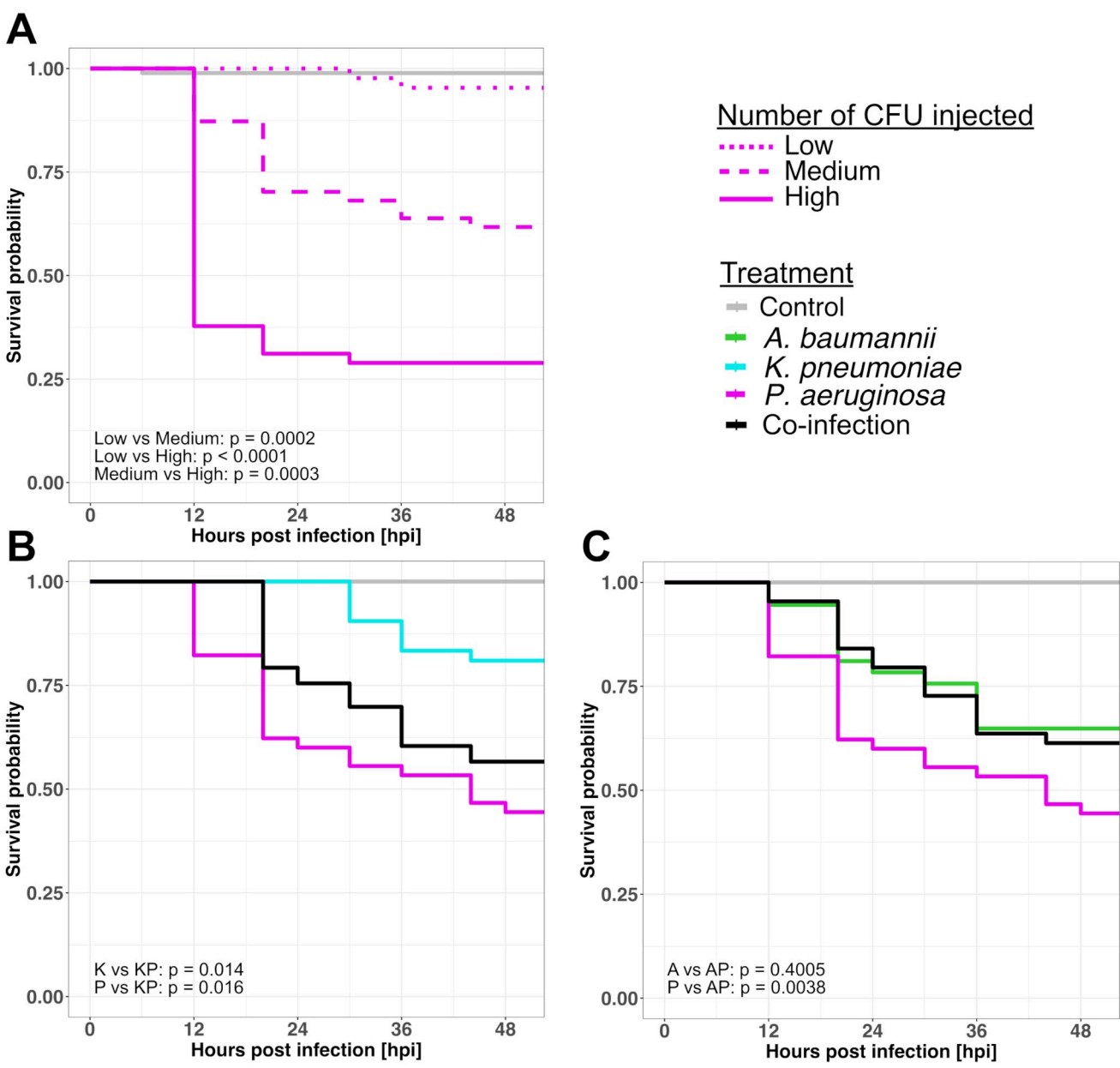

**Fig 2. Kaplan Meier survival curves of zebrafish larvae with mono- or co-infections.** (A) *P. aeruginosa* shows dose-dependent killing of zebrafish, thus confirming that the infection protocol works. The three infection doses were: low (dotted line) = 403 CFU (mean number of colony forming units), medium (dashed line) = 7625 CFU, high (solid line) = 19379 CFU. (B) and (C) Survival of zebrafish with co-infections (black curve) and the corresponding mono-infections (colored curves). For the co-infections shown in (B), the mean number of CFU is 6850 for *K. pneumoniae* and 5750 for *P. aeruginosa*. For the co-infections shown in (C), the mean number of CFU is 8600 for *A. baumannii* and 5750 for *P. aeruginosa*. All zebrafish embryos were infected 2 days post-fertilization and survival (y-axis) was subsequently monitored every 4–8 h for 54 h in total (x-axis). Control larvae were injected with a 0.8% NaCl solution. We conducted multiple pairwise comparisons using the log-rank test for all panels (alpha = 0.05, Benjamini-Hochberg p-value adjustment). Data shown in (A), (B), and (C) are from two independent experiments with the following numbers of individual zebrafish per treatment: control, N = 48; (A): *P. aeruginosa* low, N = 43; medium, N = 47; high, N = 45; (B): *P. aeruginosa* mono, N = 45; *K. pneumoniae* mono, N = 42; mix, N = 53; (C): *P. aeruginosa* mono, N = 45; *A. baumannii* mono, N = 37; mix, N = 44.

variation across individuals. (i) Area occupied in the otic vesicle. While pathogens occupied a large area ($> 75\%$) of the otic vesicle in many fish, occupancy rate was much lower in certain individuals. (ii) Species frequency. The relative occupancy of the co-infecting species can vary between individual zebrafish. For example, for the *K. pneumoniae* and *P. aeruginosa* co-

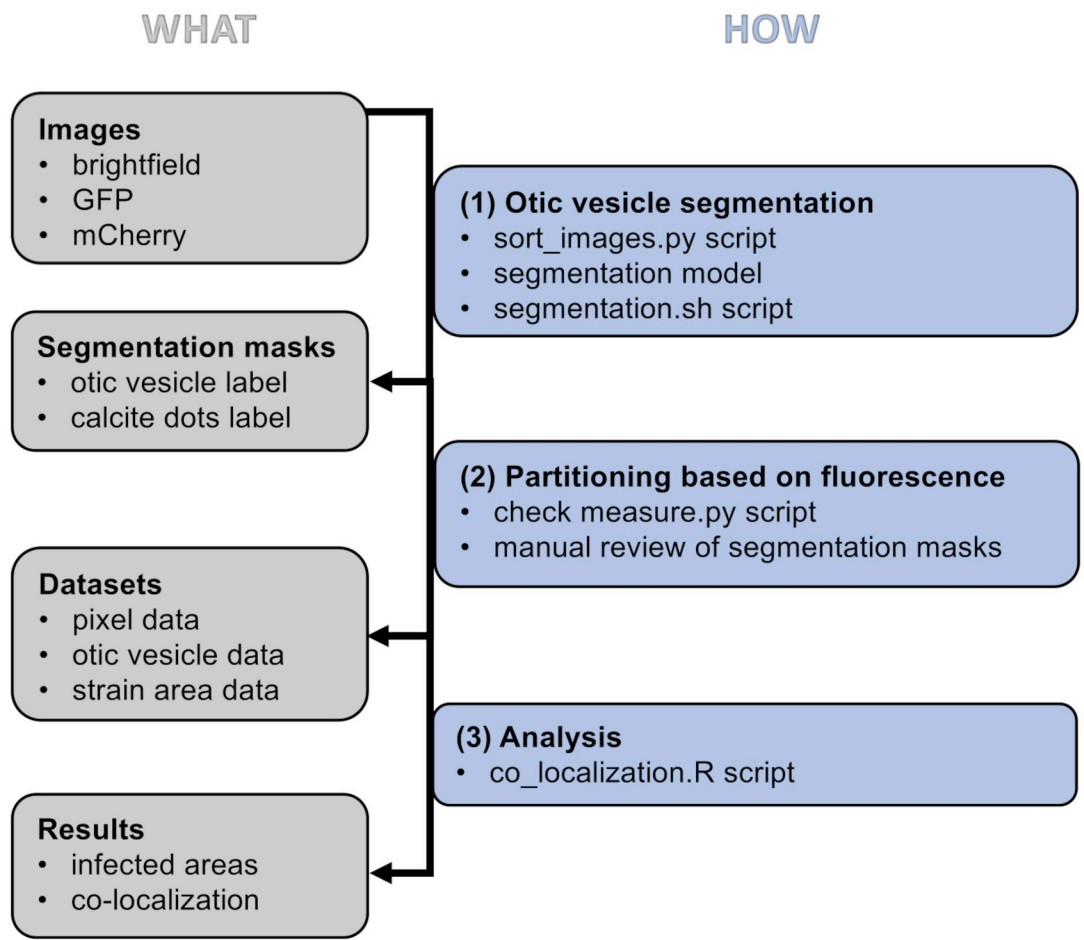

**Fig 3. Overview of the image analysis workflow of the otic vesicle (OV), the inner ear structure of the zebrafish.** The left side lists the images and datasets acquired during the workflow while the right side highlights the analysis steps and the required scripts. The three analysis steps involve: (1) Organizing raw images, segmenting the OV with the automated segmentation model based on deep learning, and collecting segmentation masks. For a Windows operating system, use the segmentation.ps1 instead of the segmentation.sh script (MacOS/Ubuntu). (2) Manually reviewing the segmentation masks, excluding low-quality segmentations, and measuring the fluorescence within the OV. (3) Visualizing and calculating co-localization of the differently tagged strains.

infection we find some fish in which *P. aeruginosa* dominates and others in which *K. pneumoniae* dominates. (iii) Pathogen co-occurrence patterns. The co-localization pattern of two co-infecting species can vary, whereby the two species completely co-localize in some fish, but spatially segregate more clearly in other individuals (Fig 4 and S2 Fig). These results show that our image analysis protocol allows both to track common patterns across individuals and to identify biological sources of variation between individuals.

## Challenges and limitations of the infection protocol

One challenge with zebrafish infections is the difficulty of accurately controlling the number of colony-forming units (CFU) injected. While cultures can be precisely adjusted to the same optical density, there can still be considerable variation in the actual number of cells injected into the zebrafish. The microloader tip is one potential source for this variation. It is used to aspirate and release the bacterial solution into the glass needle. Due to its thinness, the bacterial

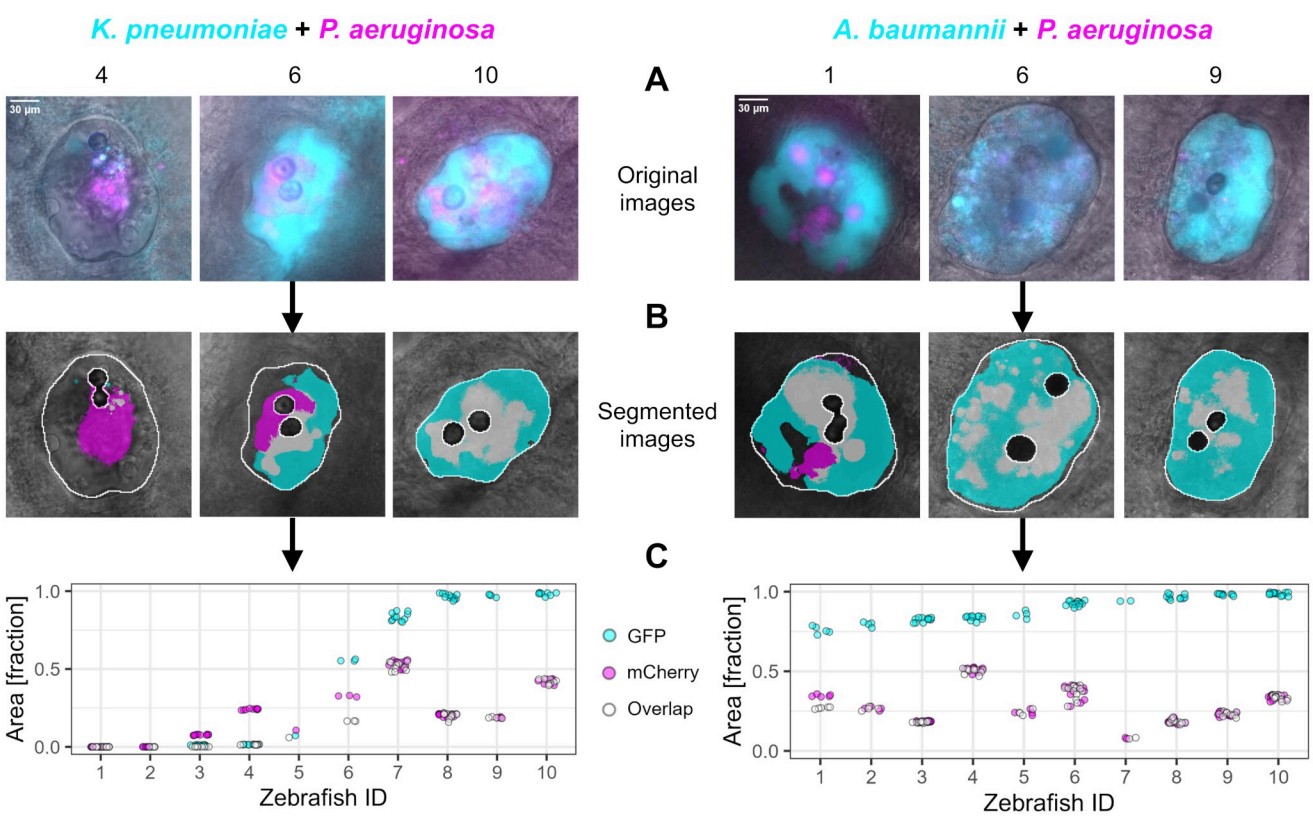

**Fig 4. The two co-infecting species can reliably be detected and quantified in the otic vesicle of zebrafish.** (A) Representative images of three individual zebrafish per pathogen combination are shown as overlays of the brightfield, GFP (excitation at 475 nm & emission at 520 nm; shown in cyan), and mCherry (excitation at 555 nm & emission at 605 nm; shown in magenta) image. (B) Masks obtained from automated segmentation are shown in white. The partitioning of the bacterial species is done using the fluorescence signal. Since the zebrafish show some inherent auto-fluorescence, bacterial occupation is defined by a fluorescence value that is twice as high as the fluorescence value observed in the surrounding tissue of the otic vesicle (a layer of approximately 16 μm, corresponding to 50 pixels). (C) Quantitative image analysis showing the relative area of the otic vesicle that is occupied by the co-infecting pathogens (y-axis) across zebrafish individuals (x-axis), ordered from lowest to highest bacterial occupation. Magenta and cyan fractions respectively represent the area occupied by either *P. aeruginosa* or the co-infecting pathogen (*K. pneumoniae* or *A. baumannii*). The mean number of CFU injected is 11800 for *K. pneumoniae*, and 12200 for *A. baumannii*. For *P. aeruginosa*, the mean number of CFU injected in the co-infection with *K. pneumoniae* is 7900, and with *A. baumannii* 11600. The grey fraction (open circles) represents the area simultaneously occupied by both pathogens. Each data point represents a z-slice imaged from the respective zebrafish ID. The data shown are from three individual experiments, with N = 10 for each co-infection. Mono-infections of each pathogen are shown in the S3 Fig.

load aspirated or released might vary considerably each time. The thin glass needle through which the bacteria are injected into the host is another potential source of variation, as the actual number of bacteria delivered into the host can vary considerably. A large number of pre-experiments (as training sessions), in which the number of CFU per species is examined, helps to reduce variation. Moreover, it is useful to define acceptable boundaries of variation. For example, for the data represented in this study, we only used experiments for which the variation in the number of CFU was below one order of magnitude: maximal variation for single infections = 29%, for co-infections = 42%.

There is an upper limit to how many CFU can be injected through the needle into the fish with high confidence because of two potential issues. First, a high bacterial load can clog the outlet of the glass needle so that no or only a few bacteria end up in the host. Second, dense bacterial solutions tend to leak out of the needle as soon as they come in touch with a liquid, such that dosed injections are no longer possible. For example, these problems arose when we

aimed at injecting about 20000–25000 CFU of *K. pneumoniae* into zebrafish larvae. We thus advise injecting fewer cells or adjusting injection and/or back pressure so that leakage is not an issue.

Another challenge is to keep injections contained within the otic vesicle. As explained in Protocol (A), the use of phenol red helps to monitor whether the injection was done correctly and stays contained. Leakage of phenol red into the surrounding tissue indicates an imprecise injection, and we recommend removing those individuals from the experiment.

In conclusion, we present (A) a protocol to establish single and multispecies bacterial infections in the inner ear structure, the otic vesicle, of the zebrafish, (B) methods for embedding zebrafish and imaging the otic vesicle, and (C) a novel toolbox to automatically segment the otic vesicle and a streamlined image analysis pipeline. The outputs of our protocols (fluorescence values of each species, overlapping fluorescence/area of species, host survival) can be used to answer various questions both on the pathogen and the host side. For example, fitness and growth of multiple pathogens can be measured over time within the host by quantifying the area occupied. Moreover, contact zones between pathogens can be assessed and pathogen-related metrics can be linked to host survival. Our protocols can further be used to study pathogen and host aspects other than those presented in our paper. For instance, the use of transgenic zebrafish lines with fluorescently labeled macrophages or neutrophils would allow the quantification of host responses to pathogens, while the use of gene-expression reporters would allow to measure bacterial responses to co-infections and host immune factors. Taken together, we are confident that the set of comprehensive protocols presented in this paper will enable researchers to explore pathogen-pathogen and pathogen-host interactions in the zebrafish as a model host to advance our understanding of polymicrobial infections.

## Supporting information

**S1 File. A new protocol for multispecies bacterial infections in zebrafish and their monitoring through automated image analysis.** Also available on protocols.io.
(PDF)

**S2 File. Protocol A: Zebrafish infections into the otic vesicle.** Also available on protocols.io.
(PDF)

**S3 File. Protocol B: Zebrafish embedding and imaging.** Also available on protocols.io.
(PDF)

**S4 File. Protocol C: Automated segmentation of the otic vesicle and image analysis.** Also available on protocols.io.
(PDF)

**S5 File. Statistical analyses.**
(XLSX)

**S6 File. Image analysis scripts.**
(ZIP)

**S1 Fig. Kaplan Meier survival curves of 2 days post-fertilization zebrafish larvae with mono- and co-infections.** Survival of zebrafish with co-infections (black curve) and the corresponding mono-infections (colored curves). The mean number of CFU is 6850 for *K. pneumoniae* and 8600 for *A. baumannii*. Survival (y-axis) was monitored every 4–8 h for 54 h in total (x-axis). Control larvae were injected with a 0.8% NaCl solution. We conducted multiple pairwise comparisons using the log-rank test (alpha = 0.05, Benjamini-Hochberg p-value

adjustment). Data are from two independent experiments with the following numbers of individual zebrafish per treatment: control, N = 48; *A. baumannii* mono, N = 37; *K. pneumoniae* mono, N = 42; mix, N = 46.
(PDF)

**S2 Fig. The two co-infecting species *A. baumannii* and *K. pneumoniae* can reliably be detected and quantified in the otic vesicle of zebrafish.** (A) Representative images of three individual zebrafish are shown as overlays of the brightfield, GFP (excitation at 475 nm & emission at 520 nm; shown in cyan), and mCherry (excitation at 555 nm & emission at 605 nm; shown in magenta) image. (B) Masks obtained from automated segmentation are shown in white. The partitioning of the bacterial strains is done using the fluorescence signal. Since the zebrafish show some inherent auto-fluorescence, bacterial occupation is defined by a fluorescence value that is twice as high as the fluorescence value observed in the surrounding tissue of the otic vesicle (a layer of approximately 16 μm, corresponding to 50 pixels). (C) Quantitative image analysis showing the relative area of the otic vesicle that is occupied by the co-infecting pathogens (y-axis) across zebrafish individuals (x-axis), ordered from lowest to highest bacterial occupation. Magenta and cyan fractions respectively represent the area occupied by either *K. pneumoniae* or *A. baumannii*, with the mean number of CFU injected being 8100 and 7400, respectively. The grey fraction (open circles) represents the area simultaneously occupied by both pathogens. Each data point represents a z-slice imaged from the respective zebrafish ID. The data shown are from one experiment with n = 10.
(PDF)

**S3 Fig. A mono-infection can reliably be detected and quantified in the otic vesicle of zebrafish.** (A) Representative images of individual zebrafish per pathogen are shown as overlays of the brightfield, GFP (excitation at 475 nm & emission at 520 nm; shown in cyan), and mCherry (excitation at 555 nm & emission at 605 nm; shown in magenta) image. (B) Masks obtained from automated segmentation are shown in white. Since the zebrafish show some inherent auto-fluorescence, bacterial occupation is defined by a fluorescence value that is twice as high as the fluorescence value observed in the surrounding tissue of the otic vesicle (a layer of approximately 16 μm, corresponding to 50 pixels). (C) Quantitative image analysis showing the relative area of the otic vesicle that is occupied by a pathogen (y-axis) across zebrafish individuals (x-axis), ordered from lowest to highest bacterial occupation. The GFP signal is shown in cyan (*A. baumannii* and *K. pneumoniae* on left) and the mCherry signal in magenta (*P. aeruginosa* and *K. pneumoniae* on right). The mean number of CFU injected from left to right is as follows: 7400, 9200, 13200, and 8100. The grey fraction (open circles) represents the area when both signals overlap. Each data point represents a z-slice imaged from the respective zebrafish ID. The data shown are from two individual experiments, with n = 5 for each mono-infection.
(PDF)

## Acknowledgments

We are thankful to Stephan Neuhauss, Kara Kristiansen, Martin Walther, Marco Garbelli, and Nicolas Rieser for their generous contribution of zebrafish as well as helpful expertise regarding any questions related to breeding and handling zebrafish. We thank Felix Ellett and Daniel Irimia for kindly sending us a rubber mold of the microstructured surface array they designed for infecting 2-day old zebrafish. Their device is available for purchase at BioMEMS Core at the Massachusetts General Hospital (https://researchcores.partners.org/biomem/about). Lastly, we thank the Center for Microscopy and Image Analysis at UZH and in particular

Johannes Riemann for the exceptional support. Fig 1 was created using BioRender (www. biorender.com).

## Author Contributions

**Conceptualization:** Désirée A. Schmitz, Rolf Kümmerli.

**Data curation:** Désirée A. Schmitz.

**Formal analysis:** Désirée A. Schmitz, Tobias Wechsler, Hongwei Bran Li.

**Funding acquisition:** Rolf Kümmerli.

**Investigation:** Désirée A. Schmitz.

**Methodology:** Désirée A. Schmitz, Tobias Wechsler, Hongwei Bran Li, Rolf Kümmerli.

**Project administration:** Désirée A. Schmitz.

**Resources:** Rolf Kümmerli.

**Software:** Tobias Wechsler, Hongwei Bran Li.

**Supervision:** Désirée A. Schmitz, Bjoern H. Menze, Rolf Kümmerli.

**Validation:** Désirée A. Schmitz, Tobias Wechsler.

**Visualization:** Désirée A. Schmitz, Tobias Wechsler, Hongwei Bran Li.

**Writing – original draft:** Désirée A. Schmitz, Tobias Wechsler, Hongwei Bran Li, Rolf Kümmerli.

**Writing – review & editing:** Désirée A. Schmitz, Tobias Wechsler, Hongwei Bran Li, Bjoern H. Menze, Rolf Kümmerli.

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
