## [Decision Letter · Decision Letter 0]

22 Mar 2024

PONE-D-24-02113A new protocol for multispecies bacterial infections in zebrafish and their monitoring through automated image analysisPLOS ONE

Dear Dr. Schmitz,

Thank you for submitting your manuscript to PLOS ONE. After careful consideration, we feel that it has merit but does not fully meet PLOS ONE’s publication criteria as it currently stands. Therefore, we invite you to submit a revised version of the manuscript that addresses the points raised during the review process.

We look forward to receiving your revised manuscript.

Kind regards,

Michael Schubert

Academic Editor

PLOS ONE

“Swiss National Science Foundation (grant no. 31003A_182499 and 310030_212266 to RK).”

3. Please note that your Data Availability Statement is currently missing the DOI/accession number of each dataset OR a direct link to access each database. If your manuscript is accepted for publication, you will be asked to provide these details on a very short timeline. We therefore suggest that you provide this information now, though we will not hold up the peer review process if you are unable.

4. We note that Figure 1 in your submission contain copyrighted images. All PLOS content is published under the Creative Commons Attribution License (CC BY 4.0), which means that the manuscript, images, and Supporting Information files will be freely available online, and any third party is permitted to access, download, copy, distribute, and use these materials in any way, even commercially, with proper attribution. For more information, see our copyright guidelines: http://journals.plos.org/plosone/s/licenses-and-copyright.

5. We note you have not yet provided a protocols.io PDF version of your protocol and/or a protocols.io DOI. When you submit your revision, please provide a PDF version of your protocol as generated by protocols.io (the file will have the protocols.io logo in the upper right corner of the first page) as a Supporting Information file. The filename should be S1_file.pdf, and you should enter “S1 File” into the Description field. Any additional protocols should be numbered S2, S3, and so on. Please also follow the instructions for Supporting Information captions [https://journals.plos.org/plosone/s/supporting-information#loc-captions]. The title in the caption should read: “Step-by-step protocol, also available on protocols.io.”

Please assign your protocol a protocols.io DOI, if you have not already done so, and include the following line in the Materials and Methods section of your manuscript: “The protocol described in this peer-reviewed article is published on protocols.io (https://dx.doi.org/10.17504/protocols.io.[...]) and is included for printing purposes as S1 File.” You should also supply the DOI in the Protocols.io DOI field of the submission form when you submit your revision.

If you have not yet uploaded your protocol to protocols.io, you are invited to use the platform’s protocol entry service [https://www.protocols.io/we-enter-protocols] for doing so, at no charge. Through this service, the team at protocols.io will enter your protocol for you and format it in a way that takes advantage of the platform’s features. When submitting your protocol to the protocol entry service please include the customer code PLOS2022 in the Note field and indicate that your protocol is associated with a PLOS ONE Lab Protocol Submission. You should also include the title and manuscript number of your PLOS ONE submission.

Reviewers' comments:

Reviewer's Responses to Questions

**Comments to the Author**

1. Does the manuscript report a protocol which is of utility to the research community and adds value to the published literature?

Reviewer #1: No

Reviewer #2: Yes

Reviewer #3: Yes

2. Has the protocol been described in sufficient detail?

To answer this question, please click the link to protocols.io in the Materials and Methods section of the manuscript (if a link has been provided) or consult the step-by-step protocol in the Supporting Information files.

The step-by-step protocol should contain sufficient detail for another researcher to be able to reproduce all experiments and analyses.

Reviewer #1: No

Reviewer #2: Partly

Reviewer #3: Yes

3. Does the protocol describe a validated method?

Reviewer #1: No

Reviewer #2: Yes

Reviewer #3: Yes

4. If the manuscript contains new data, have the authors made this data fully available?

Reviewer #1: Yes

Reviewer #2: Yes

Reviewer #3: Yes

5. Is the article presented in an intelligible fashion and written in standard English?

Reviewer #1: Yes

Reviewer #2: Yes

Reviewer #3: Yes

6. Review Comments to the Author

Reviewer #1: In this study, the Authors described the protocol they develop for establishing and tracking single and multispecies bacterial infections in the otic vesicle of the zebrafish by imaging. They also generate an image analysis pipeline for the automated segmentation of this structure, and quantify pathogen load through fluorescence intensity.

Essentially, this study is based on the description of a new methodology developed in a zebrafish model of bacterial co-infection. In my opinion, the proposed methodology is not as innovative or faster or more easily usable compared to existing techniques. Therefore, the proposed work, although the imaging analysis technology described is novel, does not present sufficiently innovative details is more suitable for a methodological journal.

Please see attached file.

Reviewer #2: Review for manuscript by Schmitz DA et al titled “A new protocol for multispecies bacterial 1 infections in zebrafish and their 2 monitoring through automated image analysis.”

The authors described a novel method to investigate multispecies bacterial infections in a zebrafish model. Zebrafish is a very upcoming model system for infectious diseases but previous studies are only investigated monospecies infection. For that reason this study is very important and will provide a lot of opportunities to study other multispecies infections. The development of a publicly available image analysis pipeline is a very valuable addition to the study.

The manuscript is well-written, the experimental approach needs some clarifications. In the supplementary materials there are 3 very well described, detailed methodology which will be greatly appreciated for other scientists.

A few required corrections/ clarification:

1. In the introduction: “Nevertheless, they already have innate immunity.” Please provide references

2. It was stated that zebrafish was infected with increasing amounts P. aeruginosa in two independent experiments – why not 3 independent experiments and what were the Ns of the other mono and multi species infection experiments?

3. Figures are blurry, please work on them.

Reviewer #3: In this manuscript the authors describe a new larval zebrafish inner ear model to assess infections by multiple pathogens at the same time. The have developed AI-based screens to automate the monitoring of these multiple pathogens within the ears of these fish. Imaging strategies are also described. The protocols are nicely detailed and should be easy for readers to follow.

The ms is well written but could use some editing for English grammar.

Specific points:

1) Figure 2: what is “survival probability?” Either they are dead or not at the specific time points so this makes no sense.

7. PLOS authors have the option to publish the peer review history of their article (what does this mean?). If published, this will include your full peer review and any attached files.

Reviewer #1: No

Reviewer #2: **Yes: **Eva Sapi Ph.D.

Reviewer #3: No

---

## [Author Response · Author response to Decision Letter 0]

4 May 2024

All responses to editors and reviewers have been uploaded in a separate letter in a file labeled 'Response to Reviewers'.

---

## [Decision Letter · Decision Letter 1]

20 May 2024

A new protocol for multispecies bacterial infections in zebrafish and their monitoring through automated image analysis

PONE-D-24-02113R1

Dear Dr. Schmitz,

We’re pleased to inform you that your manuscript has been judged scientifically suitable for publication and will be formally accepted for publication once it meets all outstanding technical requirements.

Kind regards,

Michael Schubert

Academic Editor

PLOS ONE

Additional Editor Comments (optional):

Please take note of the reviewer comment concerning the lack of image sharpness.

Reviewers' comments:

Reviewer's Responses to Questions

**Comments to the Author**

1. Does the manuscript report a protocol which is of utility to the research community and adds value to the published literature?

Reviewer #1: Yes

Reviewer #2: Yes

Reviewer #3: Yes

2. Has the protocol been described in sufficient detail?

To answer this question, please click the link to protocols.io in the Materials and Methods section of the manuscript (if a link has been provided) or consult the step-by-step protocol in the Supporting Information files.

The step-by-step protocol should contain sufficient detail for another researcher to be able to reproduce all experiments and analyses.

Reviewer #1: Yes

Reviewer #2: Yes

Reviewer #3: Yes

3. Does the protocol describe a validated method?

Reviewer #1: Yes

Reviewer #2: Yes

Reviewer #3: Yes

4. If the manuscript contains new data, have the authors made this data fully available?

Reviewer #1: Yes

Reviewer #2: Yes

Reviewer #3: N/A

**5. Is the article presented in an intelligible fashion and written in standard English?**

Reviewer #1: Yes

Reviewer #2: Yes

Reviewer #3: Yes

6. Review Comments to the Author

Reviewer #1: I have reviewed the supplementary materials that were not accessible for my previous revision due to technical problems (as discussed with plos one supporting staff). Now the manuscript is suitable for publication in the Plos one lab protocol section)

Reviewer #2: The authors addressed all concerns by the 3 reviewers. The only thing I still would like to see sharper images.

Reviewer #3: All concerns were addressed. Nothing more to add here. Why does this have a minimum character requirement for a resubmission? xxxxxxxxxxxxxxxxxxxxxxxxxxxxxxxxxxxxxxxxxxxxxxxxxxxxxxxxxxxxxxxxxxxxxxxxxxxxxxxxxxxxxxxxxxxxxxxxxxxxxxxxxxxxxxxxxxxxxxxxxxxxxxxxxxxxxxxxxxxxxxxxxxxxxxxxxxxxxxxxxxxxxxxxxxxxxxxxxxxxxxxxxxxxxxxxxxxxxxxxxxxxxxxxxxxxxxxxxxxxxxxxxxxxxxxxxxxxxxxxxxxxxxxxxxxxxxxxxxxxxxxxxxxxxxxxxxxxxxxxxxxxxxxxxxxxxxxxxx

7. PLOS authors have the option to publish the peer review history of their article (what does this mean?). If published, this will include your full peer review and any attached files.

Reviewer #1: No

Reviewer #2: **Yes: **Eva Sapi

Reviewer #3: No
